# Do We Really Need Labels for Backdoor Defense?

## Abstract

Since training a model from scratch always requires massive computational resources recently, it has become popular to download pre-trained backbones from third-party platforms and deploy them in various downstream tasks. While providing some convenience, it also introduces potential security risks like backdoor attacks, which lead to target misclassification for any input image with a specifically defined trigger (*i.e.*, backdoored examples). Current backdoor defense methods always rely on clean labeled data, which indicates that safely deploying the pre-trained model in downstream tasks still demands these costly or hard-to-obtain labels. In this paper, we focus on how to purify a backdoored backbone with only unlabeled data. To evoke the backdoor patterns without labels, we propose to leverage the unsupervised contrastive loss to search for backdoors in the feature space. Surprisingly, we find that we can mimic backdoored examples with adversarial examples crafted by contrastive loss, and erase them with adversarial finetuning. Thus, we name our method as Contrastive Backdoor Defense (CBD). Against several backdoored backbones from both supervised and self-supervised learning, extensive experiments demonstrate our unsupervised method achieves comparable or even better defense compared to these supervised backdoor defense methods. Thus, our method allows practitioners to safely deploy pre-trained backbones on downstream tasks without extra labeling costs.

## 1 Introduction

While deep neural networks (DNNs) have achieved promising performance on various tasks, including computer vision (He et al., 2016) and natural language processing (Floridi & Chiriatti, 2020), their success heavily relies on a huge amount of data, massive computational resources, and carefully tuning of hyper-parameters. Thus, it becomes popular to download a pre-trained backbone and deploy it on several downstream tasks in recent years (Newell & Deng, 2020; Tan et al., 2018; He et al., 2019). These backbones can be trained in any training paradigms, including supervised learning and self-supervised learning (Chen et al., 2020; He et al., 2022; Gidaris et al., 2018), and then be open-sourced on third-party platforms.

While providing convenience, they also bring potential risks such as backdoor attacks. Numerous works (Gu et al., 2017; Nguyen & Tran, 2021; Turner et al., 2019) pointed out this threat easily occurs in supervised learning, and recent studies (Saha et al., 2022; Jia et al., 2022) started to pay attention to backdoor attacks in self-supervised learning. Specifically, a backdoored DNN always predicts a predefined label for any input image with a specific trigger. For example, a traffic sign recognition system based on a backdoored backbone may always predict the "STOP" sign as "GO STRAIGHT" in the presence of a specific pattern, which causes severe security problems.

To address this security issue, many defense methods (Zeng et al., 2022; Wang et al., 2019; Wu & Wang, 2021) are proposed. Unfortunately, almost all methods focus on backdoor attacks inside supervised learning DNNs by building a classification-based loss to defend. In the popular deployment scheme from the pre-trained backbone to downstream tasks, the practitioners might have few costly labeled data, fail to obtain a classifier head (*e.g.*, a self-supervised backbone) to compare with the true label, or feel difficult to design a classification-based loss (*e.g.*, tasks for detection or segmentation). To break through these restrictions, we first consider the following question:

*Do we really need labels for backdoor defense?*

In this paper, we focus on how to purify a backdoored backbone with only unlabeled data. Regarding the backdoor trigger as a "shortcut" (Wang et al., 2019) in decision boundary (a small trigger is enough to change outputs for many backdoored models), the traditional methods (Wang et al., 2019; Zeng et al., 2022) attempt to make the prediction deviate from the ground-truth label as far as possible using a small perturbation in inputs, so as to evoke the backdoor behavior and then erase it. Unfortunately, we have no access to any labels, and even the prediction results if the backbone lacks a classifier head. To evoke the backdoor behavior without labels, we propose to leverage the unsupervised contrastive loss to search for the backdoor in the feature space, *i.e.*, we try to make the output feature as different from its original feature as possible using a small perturbation. Surprisingly, we find that we can easily mimic backdoored examples with adversarial examples crafted by contrastive loss. Based on this finding, we propose to erase the backdoor behaviors by letting these contrastive loss-based adversarial have similar features as their clean counterpart using fine-tuning. Thus, we term our method as Contrastive Backdoor Defense (CBD), which successfully defends against backdoor attacks without any labeled data. Our main contributions are summarized as follows,

- We explore a more practical backdoor defense that requires no access to labeled data or the classifier head. It is quite suitable in the recently popular case in which the practitioner downloads a pre-trained backbone and then deploys it in the downstream tasks.

- We find that adversarial samples generated by the contrastive loss approach the cluster of backdoor samples in the hidden feature space. Inspired by it, we introduce a fine-tuning based method that can purify the backdoored backbone without any labeled data.

- We conduct comprehensive experiments to verify the effectiveness of our method across different datasets and backdoor attacks. Empirically, our unsupervised method achieves comparable or even better defense compared to previous supervised defense.

## 2 RELATED WORK

**Backdoor Attack.** Backdoor Attack is a newly risen security concern on DNNs (Gu et al., 2017), in which the adversary can manipulate the model to predict a target class as long as a predefined trigger pattern appears in the image. This backdoor behavior can be easily injected inside DNNs by poisoning some data pairs. Specifically, (1) **poison-label attack**: the attacker randomly adds the trigger pattern into samples from all classes and changes their label to the target class (Gu et al., 2017; Chen et al., 2017; Nguyen & Tran, 2021; Zbontar et al., 2021; Doan et al., 2021)). (2) **clean-label attack**: the adversary only adds the trigger pattern into the samples from the target class, which is more stealthy since their annotation is correct (Turner et al., 2019). Recent studies start to pay attention to backdoor attacks on self-supervised learning frameworks, especially on contrastive learning methods (Saha et al., 2022; Jia et al., 2022). This emerging threat is challenging for DNN models and attracts researchers' attention.

**Backdoor Defense.** Meanwhile, numerous defense methods are proposed, which can be mainly grouped into two categories, including (1) **training-time defense** (Li et al., 2021a; Huang et al., 2022; Gao et al., 2021): the defender can access training data and train a model based on various defense strategies. For instance, Gao et al. (2021) utilized adversarial training to train a robust model against backdoor triggers; (2) **post-processing defense** (Liu et al., 2018; Wang et al., 2019; Wu & Wang, 2021; Zeng et al., 2022; Li et al., 2021b): the defender sanitize the models with tiny amounts of data with no access to the training process and training data. Thus, post-processing defense can be applied in a wider range of scenarios, *e.g.*, purifying backbones from the Internet before deploying them in downstream tasks. However, almost all these methods rely on enough amount of labeled clean data and classification loss, while labeled data may be hard to obtain, the backbone may have no classifier head, or it is hard to design classification-based loss for defense (*e.g.*, defense for object detection or segmentation). In this work, we focus on how to purify backdoored backbone without the help of any labels.

## 3 CONTRASTIVE BACKDOOR DEFENSE WITHOUT ANY LABELED DATA

In this section, we first define a practical problem setup on pre-trained backbones. We then analyze the existing backdoor behaviors in the feature space and propose a new way to mimic these behaviors even without trigger patterns and ground-truth labels.

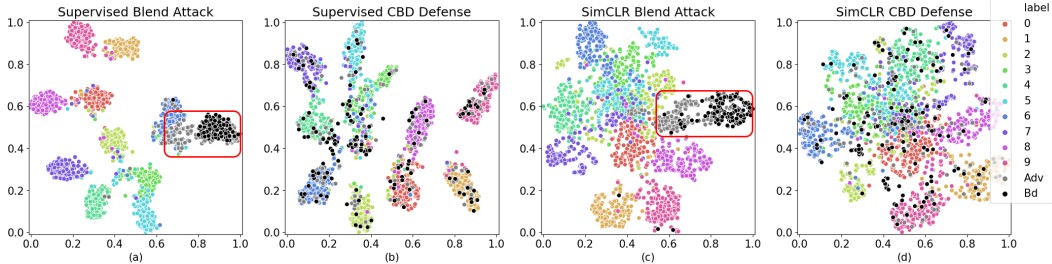

Figure 1: The t-SNE visualization of samples in feature space on CIFAR-10. All backbones are backdoored by the Blend Attack with class 6 as the target label. **(a)-(b)**: supervised backbone. **(c)-(d)**: self-supervised backbone. **(a)&(c)**: backboored backbone. **(b)&(d)**: backbone after proposed CBD. *Bd*: backdoor examples; *Adv*: contrastive adversarial examples.

## 3.1 PRELIMINARIES

**Backbone Training.** For the backbone $f(\cdot, \theta)$ from supervised learning, its parameters are usually trained based on a $K$-class classification problem. Given a labeled training dataset $\mathcal{D}_l = \{(x_1, y_1), \cdots, (x_N, y_N)\}$, which contains $N$ inputs $x_i \in \mathbb{R}^d, i = 1, \cdots, N$, and the corresponding ground-truth label $y_i \in \{1, \cdots, K\}$, the cross-entropy loss for a single data pair $(x_i, y_i)$ can be calculated as follows,

$$\ell_{ce}(x_i) = -\log g_{y_i}(f(x_i, \theta)), \tag{1}$$

where the $g(\cdot)$ is the classifier head for this classification task and $g_{y_i}(\cdot)$ indicates the outputted probability that $x_i$ belongs to class $y_i$ from the classifier. The training process attempts to find an optimal model parameter $\theta$ to minimize the average loss on the whole training data.

By contrast, the backbone $f(\cdot, \theta)$ from self-supervised learning is trained without any classifier head. The training process optimized the model parameters, so as to let the similar sample pairs stay close to each other while dissimilar ones are far apart in the embedding space. For example, given an unlabeled dataset $\mathcal{D}_u = \{x_1, \cdots, x_N\}$, the normalized temperature-scaled contrastive loss for the sample $x_i$ is

$$\ell_{cl}(\tilde{x}_i, \hat{x}_i) = -\log \frac{\exp(\text{sim}(\tilde{z}_i, \hat{z}_i)/\tau)}{\sum_{k=1}^{2N} \mathbb{I}_{[k \neq i]} \exp(\text{sim}(\tilde{z}_i, \tilde{z}_k)/\tau)}, \tag{2}$$

where $\tilde{z}_i = f(\tilde{x}_i, \theta), \hat{z}_i = f(\hat{x}_i, \theta), \tilde{z}_k = f(\tilde{x}_k, \theta)$, $\text{sim}(\cdot, \cdot)$ is the cosine similarity and $\tau$ is the temperature hyper-parameter. $\tilde{x}_i$ and $\hat{x}_i$ (positive samples) are two augmented samples from the same sample $x_i$, while $\tilde{z}_k$ (negative samples) is the projection of any other augmented samples. The self-supervised training process attempts to find an optimal model parameter $\theta$ to minimize the average loss over all possible positive pairs on the unlabeled dataset.

**Defense Setting.** Here, we consider a typical setting that one practitioner downloads a pre-trained backbone from an untrustworthy source, and defends against potential backdoor attacks before deploying it. Note that the pre-trained backbone can be trained by supervised or self-supervised methods. However, there are not any labels or label-related information at hand for defense. This is very different from existing backdoor defense methods that rely on clean labeled data.

## 3.2 BACKDOOR BEHAVIORS IN THE PRE-TRAINED BACKBONE

In this section, we first present our observation of the backdoor "shortcut" in the pre-trained backbone. Then, we provide an analysis that instance-wise adversarial examples can potentially find such "shortcut" feature, which inspires the methodology of designing backdoor defense.

**Visualization of Backdoor Attacks.** Starting from supervised learning, a successful backdoor attack misclassifies triggered samples into the target class. As shown in the 1st plot of Figure 1, the clean samples from the target class (blue circles) and the backdoor samples (black circles) are located in two separate clusters, though they are classified into the same class. For self-supervised learning, without knowing the downstream task, a successful backdoor attack can only be verified by a disparity between benign features and backdoor features (Carlini & Terzis, 2022). As shown in the 3rd plot of Figure 1, the backdoor cluster (black circles) is also obviously separated from clean clusters. This consistent phenomenon inspires a potential defense via identifying the backdoor cluster and removing it. This type of approach is easy to realize when labels are available (Wang

et al., 2019). However, when there are not any labels, we need to design new methods to identify the separated backdoor cluster.

**Covering Backdoor Cluster via Adversarial Examples.** We first bring a possible solution in a supervised manner. Previously, Wang et al. (2019) regard the backdoor trigger as a kind of "shortcut" in decision boundary, because a small trigger is enough to change outputs for the backdoored model. This provides a possible approach, *i.e.*, making the prediction of the backdoor sample deviate from its ground-truth label as far as possible using a small perturbation in inputs, so as to reconstruct this "shortcut". It is natural to connect the "shortcut" recovery process to generating adversarial perturbations, which maximize the loss of ground-truth labels in the inference phase. Other works (Shan et al., 2020; Mu et al., 2022; Weng et al., 2020; Gao et al., 2021) also explore the connection between adversarial examples and backdoored models. Unfortunately, these methods require labeled data, which might be hard to obtain or costly[1]. For a more general defense on the pre-trained backbone, situations where the classifier head or labeled data is inapplicable (*e.g.*, self-supervision backbone) should involve into consideration.

Motivated by this connection, we are speculating if we can maximize the disparity of features, so that we can generate perturbation to discover "shortcut" without labels. Recent progress of unsupervised contrastive loss become a promising tool to express this difference. Inspired by contrastive adversarial examples that only depends on unlabeled instances (Fan et al., 2021; Kim et al., 2020; Jiang et al., 2020), we attempt to make the contrastive loss between two views of a benign instance as far as possible in the hidden feature space. We try to discover a small perturbation to come across the feature gap between the clean clusters and the backdoor cluster. Specifically, our approach uses single image perturbation method to generate the adversarial perturbations. We only add perturbation on one of two augmentations in contrastive loss ($\ell_{cl}$). Suppose we have an image $x$ and a pair of augmentations $(\tilde{x}_i, \tilde{x}_j)$. To discover an potential backdoored perturbation of image $x$, we want to maximize the contrastive loss between features of augmentation $\tilde{x}_j$ and augmentation $\tilde{x}_i + \delta_i$ from backbone $f(\cdot, \theta)$ using PGD attack (Madry et al., 2018), where $\delta_i$ is the perturbation. Formally, perturbation $\delta_i$ can be gotten by iteratively optimizing:

$$\max_{\|\delta_k\|_p \leq \epsilon} \ell_{cl}(\tilde{x}_i + \delta_i, \tilde{x}_j)$$

where $\|.\|_p$ is $L_p$ norm and $\epsilon$ is perturbation budget.

As shown in the 1st plot and the 3rd plot of Figure 1 marked by the red rectangle, generated adversarial features (grey circle) in either supervised backbone (1st plot) or self-supervised backbone (3rd plot) attempt to cover the region of backdoor examples (black circles). This indicates the separated backdoor cluster in feature space can be approached by instance-wise contrastive examples.

### 3.3    THE PROPOSED METHOD

After approaching the backdoor cluster, our goal is to eliminate the "shortcut" and preserve the clean performance. Similar to our approach of instance-wise contrastive loss on adversarial examples which measure the gap of features, we will introduce a contrastive alignment defense method based on this idea. Specifically, we propose a Backdoor-to-Standard Pulling to overlap the backdoor clusters, and an Embedding Distillation loss to perform feature-wise knowledge distillation to preserve the utility of backbones. Finally, a Standard Fine-tuning loss to fine-tune and align the different clean variants.

**Backdoor-to-Standard Pulling.** We first illustrate how to mitigate the trigger-sensitive "shortcut". In supervised learning, the backdoor attack mainly builds a strong connection between the trigger and its target class (Huang et al., 2022). Thus, connecting the trigger to all the classes can effectively break the backdoor attack. However, this approach is infeasible in our setting since we do not use labeled data. Instead, we need to cover the gap between backdoored features and clean features. Based on our backdoored feature generation in the last section, we suspect those generated instances which approach the backdoor cluster can act as a substitution of backdoored images. Thus, pulling adversarial images toward their benign parts can mitigate backdoor effects as we align the potential backdoor features and clean features. Specifically, for each image $x$, in addition with $(\tilde{x}_i, \tilde{x}_j)$, we want to generate another view $\tilde{x}_k$, and find its perturbation $\delta_k$ by using augmentation pair $(\tilde{x}_i, \tilde{x}_k)$. After obtaining $\tilde{x}_k + \delta_k$, we treat it as a trigger recovered image, and our Backdoor-to-Standard Pulling will be:

$$\ell_{pull}(x) = \ell_{cl}(\text{freeze}(\tilde{x}_i), \tilde{x}_k + \delta_k).$$

---

[1]This is also the reason why the practitioners prefer to deploy a pre-trained model in downstream tasks.

It is worth noting that we freeze the feature of $\tilde{x}_i$. Since we do not want to pull features of $\tilde{x}_i$ close to those of triggered images, we treat it as an anchor.

**Embedding Distillation.** We then address how to preserve the utility of a backbone. Although Backdoor-to-Standard Pulling can effectively break the backdoor "shortcut", only forcing to overlap the backdoor cluster will hurt the clean performance as it inevitably breaks the original distribution and limits the expressiveness of backbones. Thus, merely aligning the backdoor clusters is insufficient. It is crucial to transfer clean feature distribution from our original backbone to our purified backbone. Inspired by the idea of knowledge distillation, we want to distill the embedding space across two backbones. Specifically, we are using the poisoned model as a teacher, and our current model as a student. Tian et al. (2020) proved that the idea of contrastive loss can be well applied to knowledge distillation in embedding space. Based on this idea, we propose an Embedding Distillation loss to align the clean features from the current model to our original model:

$$\ell_{kd}(x) = \ell_{cl}(\tilde{x}_i, (\tilde{x}_i, \theta^*)).$$

$\theta^*$ here is the parameter of the poisoned model, we get the feature of $(\tilde{x}_i, \theta^*)$ from $f(\tilde{x}_i, \theta^*)$. It is important to note that this distillation will not preserve the backdoored feature distribution since our given data is clean.

**Standard Fine-tuning.** Finally, to retain the benign representations close, aligning the variants of different clean augmentation would be helpful. This is similar to standard contrastive fine-tuning between benign augmentations on trained backbones. We follow the same manner as normal fine-tuning does in supervised learning. However, instead of fine-tuning with cross-entropy loss and a linear classifier, we directly utilize the contrastive loss on a backbone. Therefore, our Standard Fine-tuning will be:

$$\ell_{sft}(x) = \ell_{cl}(\tilde{x}_i, \tilde{x}_j).$$

**Overall.** The final loss on sample $x$ of our fine-tuning method is a combination of three items:

$$\ell_{total} = \lambda_1 \ell_{pull}(x) + \lambda_2 \ell_{kd}(x) + \lambda_3 \ell_{sft}(x),$$

where $\lambda_1$, $\lambda_2$, and $\lambda_3$ are positive hyper-parameters and $\lambda_1 + \lambda_2 + \lambda_3 = 1$.

## 4 EXPERIMENTS

**Datasets and DNNs.** We evaluate the performance of our method on **Cifar-10** (Krizhevsky et al., 2009) and **ImageNet-100** (Tian et al., 2020; Deng et al., 2009). We use Resnet-18 (He et al., 2016) as the backbone for both supervised and CL models and select SimCLR (Chen et al., 2020) as our CL method. On Cifar-10, we train 200 epochs for supervised backbones and 1000 epochs for SimCLR. For ImageNet-100, we train 90 epochs and 400 epochs for supervised backbones and self-supervised backbones, respectively.

**Backdoor Attack Configuration.** We use BadNets (Gu et al., 2017), Blend (Chen et al., 2017), SIG (Barni et al., 2019), WaNet (Nguyen & Tran, 2021), and CLA (Turner et al., 2019) as our attack baselines. BadNets is a patch-based attack; we apply a $3 \times 3$ checkerboard as our trigger for Cifar-10 and a $32 \times 32$ patch (Saha et al., 2022) for ImageNet-100. For the supervised setting, we put it on the top left. For the self-supervised setting, we put it in the center of the image. Blend is a global attack; we generate a $32 \times 32$ and a $224 \times 224$ Gaussian noise image to fit the size of Cifar-10 and ImageNet-100. We set the blend ratio $\alpha = 0.2$ for Blend. For SIG, we adopt the poison-label setting. We use the official code and configurations except for the poisoning ratio of WaNet. However, we only show the results of WaNet under supervised learning since it can not effectively create backdoor during our experiments on contrastive learning. For Clean Label Attack (CLA), we are using the same trigger as BadNets does. We set the target label to 6. The poisoning ratio across all supervised settings is 6% of the dataset (60% of target label 6 for CLA). To create backdoor on self-supervised models, we poison 60% of target label 6[2], which is 6% of all data for Cifar-10 and 0.6% for ImageNet-100. More details of the configurations are in Appendix A.1.

**Backdoor Defense Configuration.** For the supervised setting, we select fine-tuning and state-of-the-art model post-processing defenses, including ANP (Wu & Wang, 2021), NAD (Li et al., 2021b), and Fine-pruning (Liu et al., 2018) as our baseline methods. ANP is a pruning-based method; we use the default configuration for ANP. When evaluating ANP, we make a trade-off between

---

[2]Without any labels, this is the same as injecting backdoor to frog images for Cifar-10 and lorikeet images for ImageNet-100 on the self-supervised setup

Table 1: Results on supervised Cifar-10. Accuracy (Acc) of the clean test data, Attack Sucess Rate (ASR) on poisoned test data with target labels, and Patched Accuracy(PA) on poisoned test data with original labels. The best results of fine-tuning-based methods are in **bold**.

| Attacks | Metrics | No Defense | ANP | FP | FT | NAD | CBD |
|---------|---------|------------|-----|-----|-----|-----|-----|
| BadNets | ACC | 93.25 | 91.34 | 91.11 | **91.52** | 90.76 | 89.74 |
|         | ASR | 99.95 | 0.10 | 1.72 | 7.59 | 1.58 | **1.07** |
|         | PA | 0.06 | 89.77 | 88.89 | 85.59 | 89.19 | **89.34** |
| Blend | ACC | 94.23 | 88.94 | 91.54 | **92.38** | 89.32 | 91.81 |
|       | ASR | 100 | 37.41 | 54.02 | 98.89 | 72.97 | **4.98** |
|       | PA | 0 | 38.83 | 32.31 | 1.07 | 20.26 | **81.33** |
| SIG | ACC | 94.45 | 89.43 | 91.09 | **92.26** | 89.28 | 90.89 |
|     | ASR | 99.29 | 2.08 | 4.38 | 5.24 | **2.97** | 4.98 |
|     | PA | 0.67 | 82.57 | 80.59 | **81.11** | 78.62 | 78.88 |
| WaNet | ACC | 93.67 | 93.41 | **93.37** | 91.99 | 88.88 | 88.81 |
|       | ASR | 94.88 | 0.99 | **0.28** | 7.67 | 1.01 | 3.82 |
|       | PA | 4.94 | 92.56 | **91.49** | 83.64 | 87.08 | 85.98 |
| CLA | ACC | 87.86 | 84.13 | 81.70 | 77.60 | 73.79 | **81.72** |
|     | ASR | 99.96 | 10.62 | 4.92 | 45.58 | 4.40 | **2.22** |
|     | PA | 0.04 | 77.21 | 77.68 | 48.40 | 71.48 | **81.40** |

clean accuracy and attack success rate. Fine-tuning, NAD, and Fine-pruning are fine-tuning-based methods. We adopt the settings for these defenses from BackdoorBenchmark (Wu et al., 2022). Specifically, we fine-tune the backdoored models for 10 epochs. We also use this fine-tuned model as the teacher model for NAD and run the NAD defense for 20 epochs. In addition, we prune the Fine-pruning defense until the clean accuracy reaches a 90% tolerance ratio. We then fine-tune the pruned model with 50 epochs.

For the self-supervised setting, we only consider standard contrastive fine-tuning as our baseline. Standard contrastive fine-tuning is the special case of our method when the hyper-parameter is $\lambda_1 = 0$, $\lambda_2 = 0$, and $\lambda_3 = 1$.

For our CBD on Cifar-10, we set the default hyper-parameters to $\lambda_1 = 0.3$, $\lambda_2 = 0.5$, and $\lambda_3 = 0.2$ for both training settings. We generate adversarial examples with $\epsilon = 8$, step sizes $= 0.1$, and step $= 20$ for supervised learning and step $= 100$ for self-supervised learning. For ImageNet-100, we set the $\lambda_1 = 0.3$, $\lambda_2 = 0.7$ and use adversarial examples with $\epsilon = 16$ and step $= 20$ . We run CBD for 15 epochs. More details of defense configurations can be found in Appendix A.2.

**Evaluation metric.** To evaluate the performance of clean examples, we use the clean accuracy (ACC). For the evaluation of poisoned samples, we use two different methods. The first method is the standard attack success rate (ASR) which measures the false positive samples of the targeted class. In addition, we consider the patched accuracy (PA) which denotes the original accuracy of triggered samples as a more comprehensive demonstration of the attack. The intuition behind this measurement is some triggered samples are not classified as the targeted class despite the fact that they get poor ASR. This phenomenon occurs in self-supervised learning. We believe this is because some triggers, especially global triggers, can only ensure misleading features (Carlini & Terzis, 2022). Thus, only measuring ASR for this type of attack is insufficient. When evaluating self-supervised experiments and our CBD defense, we train a linear classifier on top of the pre-trained features with 1% of clean train data.

### 4.1 EFFECTIVENESS OF CBD DEFENSE

The effectiveness of the proposed CBD method is shown in this section. We use 5% of unlabeled data for CBD across different experiments.

**Analysis of CBD defense.** To demonstrate the effectiveness of CBD, we also present a visualization of our defense method in Figure 1. The result of our proposed CBD is in the 2nd and 4th plots. By using our method, we break the clustering of generated adversarial examples (gray circle) as well as the backdoor cluster (black circle). While aligning the backdoor examples with their benign version, we retain the discriminative feature space on both supervised and unsupervised methods with our Embedding Distillation.

**Comparing supervised backbone.** We first compare CBD with four post-processing methods under the supervised setting. We assume all the baseline methods can access the classifier head for better

Table 2: Results for pseudo-label data. We leverage the prediction of the poisoned model on clean unlabeled data to generate pseudo-labels and performance baselines on them.

| Attacks | Metrics | No Defense | ANP | FP | FT | NAD | CBD |
|---------|---------|-----------|-----|-----|-----|-----|-----|
| BadNets | ACC | 93.25 | 91.54 | 92.54 | 92.71 | **92.94** | 89.74 |
|         | ASR | 99.95 | 0.08 | 4.12 | 11.33 | 6.20 | **1.07** |
|         | PA | 0.06 | 87.49 | **89.51** | 83.98 | 88.26 | 89.34 |
| Blend   | ACC | 94.23 | 87.92 | 93.11 | **93.47** | 93.43 | 91.81 |
|         | ASR | 100 | 48.69 | 75.30 | 60.73 | 66.72 | **4.98** |
|         | PA | 0 | 31.07 | 20.84 | 27.92 | 24.01 | **81.33** |
| SIG     | ACC | 94.45 | 90.66 | **93.13** | 92.55 | 92.73 | 90.89 |
|         | ASR | 99.29 | 2.10 | 10.46 | 72.44 | 57.53 | **4.98** |
|         | PA | 0.67 | 84.98 | 77.31 | 25.78 | 39.91 | **78.88** |
| WaNet   | ACC | 93.67 | 92.03 | **93.59** | 89.94 | 91.57 | 88.81 |
|         | ASR | 94.88 | 0.69 | **0.19** | 0.96 | 0.69 | 3.82 |
|         | PA | 4.94 | 90.67 | **92.49** | 88.18 | 89.86 | 85.98 |
| CLA     | ACC | 87.86 | 84.16 | **82.90** | 80.88 | 79.7 | 81.72 |
|         | ASR | 99.96 | 2.18 | 0.64 | 2.49 | **0.28** | 2.22 |
|         | PA | 0.04 | 82.61 | **82.99** | 81.65 | 81.91 | 81.40 |

Table 3: Results for SimCLR, FT here is contrastive fine-tuning.

| Defenses | BadNets | | | Blend | | | SIG | | |
|----------|---------|-----|-----|-------|-----|-----|-----|-----|-----|
|          | Acc | ASR | PA | Acc | ASR | PA | Acc | ASR | PA |
| No Defense | 85.68 | 28.73 | 61.10 | 85.36 | 43.01 | 23.12 | 85.43 | 33.34 | 56.97 |
| FT | 64.94 | 5.20 | 53.32 | 63.83 | 3.79 | 20.93 | 66.59 | 7.97 | 47.93 |
| Our method | 81.77 | 6.70 | 76.20 | 80.07 | 3.67 | 70.69 | 80.51 | 6.64 | 66.34 |

Table 4: Results on supervised ImageNet-100.

| Attacks | Metrics | No Defense | ANP | FP | FT | NAD | CBD |
|---------|---------|-----------|-----|-----|-----|-----|-----|
| BadNets | ACC | 78.15 | 65.14 | 50.00 | 61.40 | 55.08 | **66.43** |
|         | ASR | 99.90 | 49.45 | 19.88 | 97.35 | 80.08 | **13.39** |
|         | PA | 0.10 | 41.23 | 28.97 | 85.59 | 13.21 | **59.84** |
| Blend   | ACC | 80.06 | 69.42 | 50.08 | 61.48 | 56.62 | **75.39** |
|         | ASR | 99.48 | 18.16 | 18.38 | 40.10 | 26.61 | **0.12** |
|         | PA | 0.40 | 27.66 | 14.81 | 21.15 | 16.97 | **70.77** |

Table 5: Results on SimCLR ImageNet-100.

| Defenses | BadNets | | | Blend | | |
|----------|---------|-----|-----|-------|-----|-----|
|          | Acc | ASR | PA | Acc | ASR | PA |
| No Defense | 61.52 | 38.56 | 35.45 | 59.86 | 0.02 | 40.06 |
| FT | 38.22 | 5.24 | 32.63 | 34.72 | 0.58 | 21.82 |
| Our method | 57.42 | 9.57 | 49.84 | 56.39 | 0.44 | 54.36 |

comparison. For the trade-off of getting costly labeled data, we compare these methods on 1% clean labeled data. We demonstrate the results in Table 1. Our method successfully purified different attacks even without labeled data and classifier head. The attack success rate in all cases is less than 5%. Also, the gap between the original clean performance and our purified clean performance is less than 5%. More specifically, on Blend, we are the only method that effectively defends the backdoor. Besides that, we have the best performance on CLA. On SIG and BadNets, we reach comparable results with other baselines. We achieved the lowest ASR across all fine-tuning-based methods against BadNets and even higher clean accuracy than state-of-the-art ANP and NAD on SIG. CBD still generates acceptable performance in the worst case of WaNet. However, we can adjust the weight of losses or adversarial strategies to improve its performance.

**Comparing supervised backbone with pseudo-label data** Although being attacked, a modern supervised DNN can still retain high clean accuracy. Thus, if a user wants to get a clean supervised model, instead of collecting costly labeled data, they can also leverage the classification results of the backdoored model as pseudo-labels. Under this scenario, the user can use other baseline methods on unlabeled data. We conduct experiments of other baselines on 5% unlabeled data with generated pseudo labels in Table 2. Note that this practice is only applicable when the classifier head is retained and clean accuracy is high (*e.g.*, Cifar-10). Even though the generated pseudo

Table 6: Case study for object detection on PascalVOC-2007. Clean Model is a pre-trained backbone without backdoor. Average Precision (**AP**) is the evaluation metric for object detection. Patched Average Precision (**PAP**) is the patched AP for our backdoor evaluation.

| Defenses | VOC2007 | | |
| --- | --- | --- | --- |
| | Clean Model | No Defense | Our Method |
| AP | 65.86 | 65.00 | 64.65 |
| PAP | 37.06 | 30.64 | 43.82 |

labels are mostly accurate, the existing noise of these labels deteriorates other defense baselines. In particular, Fine-tuning and NAD lost their ability to defend SIG, and the ASR of fine-tuning-based methods in most cases is increased. Moreover, despite comparable results on SIG for ANP, the patched accuracy on other attacks is decreased. These results demonstrate the weakness of using pseudo-labels to defend against backdoor attacks.

**Comparing self-supervised learning** Table 3 shows the results on self-supervised backbone. We pick SimCLR as our self-supervised learning method. It is worth noting that self-supervised methods in the real-world only leverage unlabeled data. Thus, we only demonstrate the effectiveness of our method and use contrastive fine-tuning as a comparison. All the performances of backbones are tested under a linear classifier with 1% training data. While maintaining high clean accuracy, CBD successfully defends all attacks and recovers the patched accuracy. Compared with standard contrastive fine-tuning that greatly downgrades the clean performance and achieves low patched accuracy, we verify the effectiveness of CBD under contrastive learning. More results for self-supervised backbones can be found in Appendix C.

**ImageNet-100 Experiments** To examine the effectiveness of our method in real-world datasets, we compare our method using ImageNet-100 (Tian et al., 2020; Deng et al., 2009) in Tables 4 and 5. We are comparing BadNets and Blend Attack on both supervised and self-supervised backbones using the same baselines. Since ImageNet-100 is much more complex, our proposed CBD set the $\lambda_1 = 0.3, \lambda_2 = 0.7$ and not using Standard Fine-tuning. In addition, we are leveraging the same unlabeled data to train a projection head from scratch for the stability of self-supervised backbones. The detailed configurations of the ImageNet-100 dataset are in Appendix A. For all the supervised backbones, our method achieves superior performance. By using 5% of unlabeled data, we successfully achieve the best clean accuracy, attack success rate, and patched accuracy in the real-world dataset. Also, our method purifies the self-supervised a backbones with huge improvement of Patched Accuracy while only losing less than 5% of clean accuracy. Overall, these experiments can demonstrate the capability of our proposed method when other methods failed in the real-world dataset.

Also, we provide a case study for object detection in Table 6. The evaluation metric for object detection is average precision (AP) and patched average precision (PAP). In this case study, we demonstrate the effects of backdoor attack on the backbone and improve the Patched AP with our CBD. We use the pre-trained SimCLR backbone with Blend Attack on ImageNet-100. Then, we train and test the Fast-RCNN with this backbone on PascalVOC-2007 (Everingham et al., 2009) dataset. Note that even without poisoning, Blend Attack can disturb the results of object detection. Thus, for better comparison, we also add a clean pre-trained SimCLR backbone and test it on Blend Attack. Even in the downstream object detection setting, while reaching a similar AP, we mitigate the backdoor effects and bring the Patched AP even higher to that of the clean backbone. These results point out the emergent threat of backdoored pre-trained backbone and verify the effectiveness of our method. More implementation details are in Appendix B.

## 4.2 ABLATION STUDIES

In this section, we present ablation studies on Cifar-10 supervised backbones. More results on self-supervised backbones can be found in Appendix C.

**Effects on different adversarial examples** The quality of adversarial examples that mimic backdoored features plays an important role in our method. Among different adversarial settings, PGD20 attack with $L_\infty$ norm, 0.1 step size, and $\epsilon = 8$ is the default in our supervised setting. Please note that our default settings are not the best; we have made some trade-offs. As shown in Table 7, we present the results with different budgets ($\epsilon = 2, 4, 16$), steps (PGD100), and norm ($L_2$ norm with $\epsilon = 1024$). The carefully selected results can reach state-of-the-art without any labeled data. Based on the results of these adversarial examples, they successfully reach backdoored features with instance-wise contrastive loss.

Table 7: Ablation studies with different adversarial settings on Cifar-10 supervised backbones.

| Attacks | Metrics | Before | $\epsilon = 2$ | $\epsilon = 4$ | $\epsilon = 16$ | $L_2 1024$ | PGD100 | CBD (ours) |
|---|---|---|---|---|---|---|---|---|
| | ACC | 93.25 | 89.86 | 89.72 | 89.82 | **92.16** | 88.62 | 89.74 |
| BadNets | ASR | 99.95 | 1.14 | 1.24 | **1.06** | 4.65 | 1.22 | 1.07 |
| | PA | 0.06 | 89.38 | 89.12 | **89.48** | 88.73 | 88.19 | 89.34 |
| | ACC | 94.23 | 91.22 | 91.16 | 92.21 | **93.03** | 90.99 | 91.81 |
| Blend | ASR | 100 | 10.76 | 8.50 | 3.35 | **3.12** | 4.91 | 4.98 |
| | PA | 0 | 73.93 | 76.13 | **84.29** | 82.43 | 81.90 | 81.33 |
| | ACC | 94.45 | 91.09 | 90.79 | 90.98 | **92.96** | 89.51 | 90.89 |
| SIG | ASR | 99.29 | 5.01 | **4.49** | 5.58 | 43.54 | 4.95 | 4.98 |
| | PA | 0.67 | 78.29 | **78.90** | 78.46 | 51.83 | 77.73 | 78.88 |
| | ACC | 93.67 | 89.20 | 88.33 | 88.25 | **90.51** | 86.57 | 88.81 |
| WaNet | ASR | 94.88 | 1.17 | 1.22 | 1.65 | **1.15** | 2.02 | 3.82 |
| | PA | 4.94 | 88.93 | 87.99 | 87.66 | **89.90** | 85.60 | 85.98 |
| | ACC | 87.86 | **81.92** | 81.68 | 81.72 | 84.91 | 80.77 | 81.72 |
| CLA | ASR | 99.96 | 2.14 | 2.54 | **2.07** | 38.54 | 3.61 | 2.22 |
| | PA | 0.04 | 81.54 | 80.96 | **81.56** | 56.98 | 78.11 | 81.40 |

Table 8: Ablation studies of different losses on Cifar-10 supervised backbones. SFT is Standard Fine-tuning, KD is Embedding Distillation, and Pull is Backdoor-to-Standard pulling.

| Attacks | Metrics | Before | SFT | Pull & KD | Pull & SFT | Pull | CBD (ours) |
|---|---|---|---|---|---|---|---|
| | ACC | 93.25 | 88.55 | 88.84 | 80.65 | 71.76 | **89.74** |
| BadNets | ASR | 99.95 | 12.01 | 1.18 | 3.23 | 4.18 | **1.07** |
| | PA | 0.06 | 77.83 | 88.59 | 80.08 | 71.28 | **89.34** |
| | ACC | 94.23 | 89.21 | 91.03 | 85.68 | 75.70 | **91.81** |
| Blend | ASR | 100 | 91.89 | 5.18 | 5.59 | 6.77 | **4.98** |
| | PA | 0 | 6.99 | 81.10 | 76.34 | 67.34 | **81.33** |
| | ACC | 94.45 | 89.38 | 89.78 | 84.12 | 74.39 | **90.89** |
| SIG | ASR | 99.29 | 14.00 | 5.22 | 5.50 | 7.32 | **4.98** |
| | PA | 0.67 | 69.54 | 78.40 | 70.61 | 59.13 | **78.88** |
| | ACC | 93.67 | 69.80 | 86.42 | 59.81 | 54.90 | **88.81** |
| WaNet | ASR | 94.88 | 3.91 | **1.66** | 6.20 | 5.78 | 3.82 |
| | PA | 4.94 | 69.76 | **86.37** | 58.72 | 51.70 | 85.98 |
| | ACC | 87.86 | 40.39 | 80.05 | 35.79 | 27.80 | **81.72** |
| CLA | ASR | 99.96 | 12.49 | **1.87** | 12.77 | 10.18 | 2.22 |
| | PA | 0.04 | 38.11 | 80.03 | 34.00 | 26.56 | **81.40** |

**Effects on different losses.** Our proposed CBD is composed of three losses. To verify the effectiveness of these losses, we test our methods on four different variants. Specifically, we adjust the hyper-parameters of these losses accordingly. These variants include (1) Standard Fine-tuning ($\lambda_3 = 1$), (2) Backdoor-to-Standard pulling & Embedding Distillation ($\lambda_1 = 0.5, \lambda_2 = 0.5$), (3) Backdoor-to-Standard pulling & Standard Fine-tuning ($\lambda_1 = 0.5, \lambda_3 = 0.5$), and (4) Backdoor-to-Standard pulling ($\lambda_1 = 1$). We can verify their contributions to our method on Table 8. In most cases, our default setting reaches the best performance. However, the ablation studies indicate that Standard Fine-tuning is less important for our method and is optional compared with Backdoor-to-Standard pulling and Embedding Distillation.

## 5 CONCLUSION

In this paper, we proposed a fine-tuning-based method to erase the backdoor behaviors inside pre-trained backbones. We analyzed the behaviors of backdoored examples and proposed a contrastive loss-based way to approach them, based on which, we propose the Contrastive Backdoor Defense (CBD) to remove the backdoors without needing any labeled data. Extensive experiments on different datasets with various attacks demonstrate the superiority of our proposed method on different experimental settings with only unlabeled data.

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

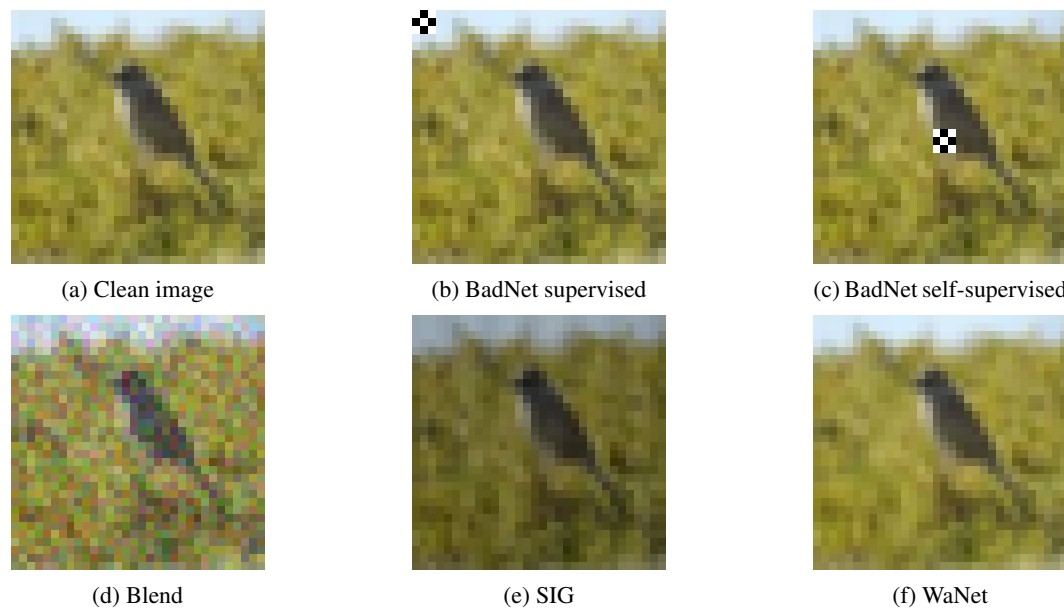

|                |                |                     |
|:--------------:|:--------------:|:-------------------:|
| (a) Clean image | (b) BadNet supervised | (c) BadNet self-supervised |
| (d) Blend | (e) SIG | (f) WaNet |

Figure 2: Examples of Cifar-10 backdoored images.

## A  MORE IMPLEMENTATION DETAILS FOR BACKDOOR ATTACKS AND DEFENSES

### A.1  BACKDOOR ATTACKS IMPLEMENTATIONS

In this part, we present the detailed configurations of our attacks. The backdoor triggers are presented in 2. We select label 6 as the target label (frog in Cifar-10 and lorikeet in ImageNet-100). We perform poison-label attack with 6% training data for supervised backbone and clean-label attack with 60% of the target category (6% of all data) for self-supervised backbone.

- **BadNets (Gu et al., 2017)**: We implant a $3 \times 3$ black-white patch for Cifar-10 and $32 \times 32$ patch introduced in (Saha et al., 2022) for ImageNet-100 as our triggers. For supervised backbones, we put the trigger on the top-left. We inject it into the center of the image to achieve a better attack success rate on self-supervised backbones.

- **Blend (Chen et al., 2017)**: We mix the data with a $32 \times 32$ and a $224 \times 224$ Gaussian noise image for Cifar-10 and ImageNet-100, respectively. We achieve good attack results in supervised and self-supervised settings with blend ratio $\alpha = 0.2$. In addition, Blend Attack can result in bad patched accuracy in self-supervised backbones even with a low attack success rate. Note that general Gaussian blur augmentation is not presented when training Blend with SimCLR.

- **SIG (Barni et al., 2019)**: We generate SIG trigger with $f = 6$ and $\Delta = 20$. We blend it to the target image with a blend ratio $\alpha = 0.3$. Note that we use poison-label attack setting for SIG in supervised learning.

- **WaNet (Nguyen & Tran, 2021)**: We use the default configurations and code of WaNet except for the poisoning ratio in supervised backbones. In particular, we poison 6% of training data and set the noise rate $p_n = 2$, $s = 0.5$, and $k = 4$. We implement WaNet based on the original code for self-supervised learning. However, we did not show the results of WaNet on self-supervised backbones as it can not build an effective backdoor with or without noise mode.

- **CLA (Turner et al., 2019)**: We are using the same $3 \times 3$ black-white patch in BadNets. To generate perturbations, we have untargeted PGD with $L_\infty, \epsilon = 16, \text{step} = 7$. Since CLA is sensitive to data augmentation, we did not use any of them in our training. This is the reason for low clean accuracy of the backdoored backbone

| Attacks | Metrics | Before | $\epsilon = 2$ | $\epsilon = 4$ | $\epsilon = 8$ | $\epsilon = 16$ | $L_2 1024$ | CBD (ours) |
|---------|---------|--------|---------------|---------------|---------------|----------------|-----------|-----------|
| | ACC | 85.68 | 81.69 | 81.91 | **81.99** | 81.90 | 81.93 | 81.77 |
| BadNets | ASR | 28.73 | 14.38 | 12.42 | 10.63 | **4.01** | 8.46 | 6.70 |
| | PA | 61.10 | 69.32 | 71.30 | 72.84 | **78.16** | 75.21 | 76.20 |
| | ACC | 85.36 | 79.45 | 79.69 | 80.25 | 80.10 | 79.97 | **80.7** |
| Blend | ASR | 43.01 | 17.27 | 12.58 | 5.26 | 2.74 | **2.23** | 3.67 |
| | PA | 23.12 | 49.67 | 57.82 | 68.28 | 72.53 | **72.67** | 70.69 |
| | ACC | 85.43 | 80.42 | 80.54 | 80.66 | **80.77** | 80.68 | 80.51 |
| SIG | ASR | 33.34 | 14.59 | 13.92 | 14.09 | 11.52 | 26.59 | **6.64** |
| | PA | 56.97 | 62.47 | 62.39 | 61.81 | 62.81 | 55.52 | **66.34** |

Table 9: Ablation studies of our methods on self-supervised backbones. The default defense for self-supervised backbone is PGD100 and PGD20 for others.

### A.2 BACKDOOR DEFENSE IMPLEMENTATIONS

To compare our results with state-of-the-art defense methods, we modify the code from open-source BackdoorBenchmark (Wu et al., 2022). To ensure the fairness of the comparison, we use 1% of labeled data and 5% of pseudo-labeled data respectively. Specifically, for pruning-based ANP (Wu & Wang, 2021), we maintain the default setting reported in the paper. We finetune a teacher model with 10 epochs and run NAD (Li et al., 2021b) defense for 20 epochs. It is worth noting that we only count the last block of ResNet-18 as the attention layer with $\beta = 1000$ as suggested in the BackdoorBenchmark. For fine-pruning (Liu et al., 2018), we also implement it with the suggested setting from the BackdoorBenchmark, where we stop pruning and finetune it with 50 epochs when the clean accuracy is dropped lower than 90% of total clean accuracy. We keep the same setting for the baselines on ImageNet-100.

For our proposed CBD on Cifar-10, we finetune the backbone with 15 epochs by using SGD optimizer with 0.02 learning rate, 0.9 momentum, and $5 \times 10^{-4}$ weight decay. Then, we attach a linear classifier to the frozen purified feature extractor and use another 1% clean data to do linear probing as suggested in other self-supervised works (Saha et al., 2022; Chen et al., 2020). We set the batch size of our defense loss to 128 and temperature to 0.5. On ImageNet-100, we use 0.05 learning rate, larger untargeted PGD with $L_\infty, \epsilon = 16, \text{step} = 20$, and keep the other parts same.

## B IMPLEMENTATION DETAILS FOR IMAGENET-100 AND OBJECT DETECTION

We describe the settings of ImageNet-100 and our object detection case study in this part. ImageNet-100 is an ImageNet subset to compare the performance of self-supervised models proposed in Tian et al. (2020). We poison 60% of the lorikeet images in ImageNet-100 (0.6% of all data) to create the backdoor. As shown in Table 6, we mitigate the backdoor effects with 5% of clean unlabeled data. Then, we test our purified backbone on object detection as a downstream task. We freeze the first two layers of our backbone and deploy it as the initial parameter of Faster-RCNN (Ren et al., 2015).

## C ABLATION STUDIES OF SELF-SUPERVISED BACKBONE

In this part, we provide ablation studies for self-supervised backbone. Similar to the supervised backbones, we conduct Cifar-10 experiments on different adversarial strategies in Table 9 and losses in Table 10. We adopt the same setting as the supervised backbone except we are using PGD100 attack to find backdoored features.

| Attacks | Metrics | Before | SFT | Pull & KD | Pull & SFT | Pull | CBD (ours) |
|---------|---------|--------|-----|-----------|-----------|------|------------|
| | ACC | 85.68 | 64.94 | 80.74 | 65.78 | 42.73 | **81.77** |
| BadNets | ASR | 28.73 | 5.20 | 6.73 | 2.27 | **0.82** | 6.70 |
| | PA | 61.10 | 53.32 | 74.83 | 59.78 | 39.56 | **76.20** |
| | ACC | 85.36 | 63.83 | 78.63 | 59.57 | 38.22 | **80.7** |
| Blend | ASR | 43.01 | 3.79 | 2.88 | **0.81** | 2.25 | 3.67 |
| | PA | 23.12 | 20.93 | **70.97** | 51.84 | 32.46 | 70.69 |
| | ACC | 85.43 | 66.59 | 79.13 | 66.48 | 45.14 | **80.51** |
| SIG | ASR | 33.34 | 7.97 | 5.42 | **1.35** | 1.99 | 6.64 |
| | PA | 56.97 | 47.93 | 65.88 | 54.42 | 40.61 | **66.34** |

Table 10: Ablation studies of different losses on self-supervised backbones.

