# OpenReview forum: "Do We Really Need Labels for Backdoor Defense?"
_ICLR.cc/2023/Conference — Submitted to ICLR 2023_

### Official Review · Reviewer_WhiY · 2022-10-22

**Confidence:** 4
**Correctness:** 3
**Technical Novelty And Significance:** 3
**Empirical Novelty And Significance:** 3
**Recommendation:** 5

**Clarity, Quality, Novelty And Reproducibility:**

The proposed idea is interesting, the paper is well written. It seems to be reproducible for the Cifar-10 dataset.

**Strength And Weaknesses:**

Advantages:
- The problem is worth studying. It could happen that the defenders have no access to the labeled data.
- The proposed framework is interesting and has proved to be effective for the Cifar-10 dataset.

Drawbacks:
- My main concern is the generation of adversarial examples. The authors propose simulating backdoored examples with adversarial examples generated by contrastive loss. This goal is unlikely to be met, in my opinion. The generated adversarial examples will typically be found outside the manifold of the attackers' training data set (either clean samples or poisoned samples via adding trigger patterns). This is especially true for real-world applications that use high-resolution images. This could be why the authors only run experiments on the Cifar-10 dataset. Working on low-resolution toy datasets does not imply that the proposed defense method can be easily applied to more difficult real-world datasets.
- The pre-trained backbone is a general backbone in a self-supervised setting (not necessarily using Cifar-10 dataset in this case). In this scenario, how did the attackers insert the backdoor for the Cifar-10 task, and what motivated them?
- In addition, the author mentioned that “To create backdoor on self-supervised model, we poison 60% of target label 6, which is 6% of all data.” Do you use the labeled data for the self-supervised training (contrastive learning) task?


**Summary Of The Paper:**

Downloading pre-trained backbones from third-party platforms and deploying them in various downstream tasks is now standard practice. This poses security risks such as backdoor attacks. The authors investigate an interesting question: Do we need labels for backdoor defense? To that end, the authors propose using unsupervised contrastive loss to look for backdoors in feature space. Backdoored examples are created by mimicking adversarial examples generated by contrastive loss. The adversarial finetuning is then used to remove the backdoors. Experiments are carried out on the Cifar-10 dataset, with the authors considering both the supervised and the CL models. The results show that the proposed defense method is effective.


**Summary Of The Review:**

Although this work has some merits, I am concerned about whether the proposed method can be applied to more difficult real-world datasets.

---

> ### Author Response · Authors · 2022-11-18
> **Response to Reviewer WhiY (2/2)**
>
> (continued)
>
> **Q3** Do you use the labeled data for the self-supervised training (contrastive learning) task?
>
> **A3:** Poisoning 60% of target label 6 is only used to demonstrate the effectiveness of our defense. This is the same as injecting backdoor on **frog** images. For simplicity, you can consider picking target label 6 as equivalent to an attacker manually picking some frog images to poison and training it on self-supervised SimCLR to mimic the potential backdoor attacks. In real practice, an attacker might release backdoored images in the same categories online, or a malicious third-party platform may intentionally poison some images and publish them. We want to demonstrate our defense in this scenario. Future works can address the problem of selecting poison images without prior knowledge of categories and attributes to achieve untargeted and unlabeled poisoning. However, our method can still tackle this attack if they still require generating backdoor cluster.
>
> [1] Saha et al., Backdoor attacks on self-supervised learning. 2022.
>
> [2] Shan et al.,  Gotta catch'em all: Using honeypots to catch adversarial attacks on neural networks. 2020.
>
> [3] Wang et al., Neural cleanse: Identifying and mitigating backdoor attacks in neural networks. 2019.
>
> ---
> Thanks for your constructive comments. Hope our explanations and additional experiments can address your concerns. Please let us know if there is more to clarify. We are happy to take your further questions during the rebuttal stage.

---

> ### Author Response · Authors · 2022-11-18
> **Response to Reviewer WhiY (1/2)**
>
> We thank Reviewer WhiY for the detailed comments on our work. We will present experiments on a real-world dataset and address other concerns point by point.
>
> ___
>
> **Q1:** It is unlikely to simulate backdoored examples with adversarial examples since adversarial examples will typically locate outside the manifold  of the training dataset (either clean samples or poisoned samples via adding trigger patterns)
>
> **A1:** We agree that adversarial examples from benign models (models trained on the clean dataset) likely locate outside the manifold of the training dataset. By contrast, the backdoored model (models trained on the poisoned dataset) has a "shortcut" in its decision boundary [2, 3], i.e., a small trigger is enough to change outputs totally. Since this is an extremely easy "shortcut" to change the labels, adversarial perturbations probably make use of it. Thus, the situation of adversarial examples on a backdoored model is different from the situation of those on a benign model in our belief. The visualization in Figure 1(a) and 1(c) also supports it, since the cluster of adversarial examples is really close to the cluster of backdoored samples.
>
> To further alleviate the concern, we also conduct experiments on ImageNet-100 with BadNets and Blend Attack as follows. Like the Cifar-10 experiments, we compare our method supervised defense baselines on supervised backbones and only compare our method with standard contrastive fine-tuning as the label is inaccessible in real-world unsupervised learning. As shown in the following table, it is clear that our method reaches the highest performance among all other baselines. More details can be found in **Section 4.1**. We can easily find that our method succeeds on both supervised and unsupervised models, which indicates we are able to use adversarial examples to mimic backdoored samples to purify backdoored models.
>
> > **Supervised backbones**
> | Attacks | Metrics | No Defense | ANP   | FP    | FT    | NAD   | CBD       |
> | ------- | ------- | ---------- | ----- | ----- | ----- | ----- | --------- |
> | BadNets | ACC     | 78.15      | 65.14 | 50.00 | 61.40 | 55.08 | **66.43** |
> |         | ASR     | 99.90      | 49.45 | 19.88 | 97.35 | 80.08 | **13.39** |
> |         | PA      | 0.10       | 41.23 | 28.97 | 85.59 | 13.21 | **59.84** |
> | Blend   | ACC     | 80.06      | 69.42 | 50.08 | 61.48 | 56.62 | **75.39** |
> |         | ASR     | 99.48      | 18.16 | 18.38 | 40.10 | 26.61 | **0.12**  |
> |         | PA      | 0.40       | 27.66 | 14.81 | 21.15 | 16.97 | **70.77** |
>
> > **Self-supervised backbones**
> | Defenses   |       | BadNets |       |       | Blend |       |
> | ---------- | ----- | ------- | ----- | ----- | ----- | ----- |
> |            | Acc   | ASR     | PA    | Acc   | ASR   | PA    |
> | No Defense | 61.52 | 38.56   | 35.45 | 59.86 | 0.02  | 40.06 |
> | FT         | 38.22 | 5.24    | 32.63 | 34.72 | 0.58  | 21.82 |
> | Our method | 57.42 | 9.57    | 49.84 | 56.39 | 0.44  | 54.36 |
>
> ---
>
> **Q2** How did the attackers insert the backdoor for a general self-supervised backbone on an irrelevant downstream task, and what motivated them?
>
> **A2** In fact, we provide a case study of a backbone pre-trained by ImageNet-100 and use an irrelevant PascalVOC-2007 dataset for an object detection downstream task in **Section 4.1**. We inject Blend Attack on the pre-trained backbone. Blend Attack is a global attack that will cover the whole image. To insert Blend Attack to this downstream task, we resize our Blend trigger to fit the size of downstream images and inject it like how we poison the backbone. We test this practice on both backdoored backbone and purified backbone. Specifically, with similar Average Precision (AP), our purified backbone increases the Patched AP (e.g., from 30.64% to 43.82%). This can show that backdoor can mislead different downstream tasks. Since the backbone in downstream tasks will be frozen or partially frozen, the deviation between backdoor samples and clean samples in the feature space will inherit to the downstream tasks, thus causing inconsistent behavior when the trigger is present.
>
> Their motivation is to cause harmful damage to real-world applications. The attacker, for this purpose, might consider confusing downstream tasks fine-tuned by a poisoned backbone. For instance, if a self-driving system is fine-tuned by a backdoored backbone, when the attacker appends a trigger to a stop sign, the system might detect it as a different sign, thus not stopping. We think the backdoor attack of the self-supervised backbone can have more extensions and different setups. We are happy to see further exploration in this area.
>
> (more below)

---

### Official Review · Reviewer_eqcZ · 2022-10-24

**Confidence:** 3
**Correctness:** 3
**Technical Novelty And Significance:** 3
**Empirical Novelty And Significance:** 3
**Recommendation:** 5

**Clarity, Quality, Novelty And Reproducibility:**

### Clarity
- The connection between adversarial and backdoor examples is not discussed. I cannot find a clear explanation on why we can mitigate backdoors by pulling adversarial images toward their benign parts.
- I guess the ImageNet-100 experiment is about image classification, but the writing is really confusing by combining it with the detection experiment description.
- Near the end of page 2: the phrase "defense for detection or detection" is hard to understand. Is it "defense for detection or segmentation" instead?
- Fig. 1: The text font is too small. It is not printing-friendly.

### Quality
The writing should be improved more, as mentioned above.

### Novelty
The proposed method seems novel and interesting. This paper also discusses defending contrastive-learning-based models, which is new.

### Reproducibility
No code is submitted, and there is no statement about code release. However, there are some descriptions of how to implement the method and run the experiments.

**Strength And Weaknesses:**

### Strengths
- The paper defines an interesting problem of mitigating backdoor effects using unlabelled data. It proposes a novel method to handle this problem that smartly employs contrastive learning.
- From the experiments, CBD shows quite competitive mitigation performance compared to the baseline backdoor mitigation methods when using labels, but more stable in the case of noisy labels.
- This paper also discusses defending contrastive-learning-based models, which is new. CBD works better than its defined counterparts when defending such models.

### Weaknesses
- The connection between adversarial and backdoor examples is not discussed. I cannot find a clear explanation on why we can mitigate backdoor by pulling adversarial images toward its benign parts.
- The experiments are conducted only on CIFAR-10. More datasets should be used.
- CBD often introduces high clean accuracy drops (3-5%). It makes the defense impractical in real life. Though, I agree that the baseline methods face the same issue.
- By the way, the results look much better using L2 norm with \epsilon = 1024. I would recommend using that configuration instead.
- I guess the ImageNet-100 experiment is about image classification, but the writing is really confusing by combining it with the detection experiment description.
- Near the end of page 2: the phrase "defense for detection or detection" is hard to understand. Is it "defense for detection or segmentation" instead?
- Fig. 1: The text font is too small. It is not printing-friendly.
- Page 4: A typo "3nd".

**Summary Of The Paper:**

This paper proposes Contrastive Backdoor Defense (CBD), a novel backdoor defense that can exploit unlabeled data to remove potential backdoors in a provided model. It first generates untargeted adversarial examples via a contrastive loss. It then proposes to finetune the model using a Backdoor-to-Standard Pulling loss that pulls adversarial images toward its benign parts, hoping it can mitigate backdoor effects as the active neurons of the backdoor are changed. That loss is used along with the standard finetuning loss and a knowledge distillation loss. CBD was tested only on CIFAR-10. It shows quite competitive mitigation performance compared to the baseline backdoor mitigation methods when using labels, but more stable in the case of noisy labels. CBD also works better than its counterparts when defending self-supervised models.

**Summary Of The Review:**

The paper defines an interesting problem of mitigating backdoor effects using unlabelled data, and proposes a novel method to handle it. CBD shows quite competitive mitigation performance compared to the baseline backdoor mitigation methods when using labels, but more stable in the case of noisy labels. It also works better than its defined counterparts when defending contrastive-learning-based models.

However, the intuition behind the adversarial-clean pulling loss is unclear and unconvincing. More datasets should be used in experiments. The results look much better using L2 norm with \epsilon = 1024, and I would recommend using that configuration instead.

---

> ### Author Response · Authors · 2022-11-18
> **Response to Reviewer eqcZ (2/2)**
>
> (continued)
>
> **Q4** The results look much better using L2 norm with \epsilon = 1024. Is it better to use this configuration instead?
>
> **A4:** We agree that the defense with $L_2$ bound sometimes performs better, while it only has unsatisfactory performance against the attack with SIG and the newly added Clean Label Attack. Since a defense method will not know which attacks it will face, it needs to perform well even in the worst case so as to secure the AI system. Thus, we choose to use defense with $L_{\infty}$, which makes ASR below 5\% even in the worst case (the following table quoted from Table 7 in the paper).
> | Attacks | Metrics | Before |$L_2$ $\epsilon=1024$ | CBD       |
> |----------|-------------|-------------|-------------|-------------|
> | BadNets | ACC   | 93.25    | 92.16 | 89.74    |
> |  | ASR     | 99.95   | **4.65**      | **1.07** |
> |  | PA        | 0.06      | 88.73       | 89.34 |
> | Blend   | ACC       | 94.23  | 93.03 | 91.81    |
> | | ASR       | 100   | **3.12** | **4.98** |
> |  | PA    | 0  | 82.43 | 81.33    |
> | SIG   | ACC       | 94.45       | 92.96    | 90.89    |
> | | ASR       | 99.29    | **43.54**           | **4.98** |
> | | PA        | 0.67    | 51.83           | 78.88 |
> | WaNet   | ACC       | 93.67    |  90.51  | 88.81    |
> | | ASR       | 94.88    |  **1.15**     | **3.82**     |
> |  | PA        | 4.94       |  89.90 | 85.98    |
> |CLA  | ACC       | 87.86      | 84.91       |  81.72 |
> |  | ASR       | 99.96    |  **38.54**       |  **3.61**    |
> |  | PA        | 0.04     | 56.98         | 81.40 |
>
>
> ---
>
> **Q5** Confusing when combining ImageNet-100 with the detection.
>
> **A5** Thanks for your suggestion, we updated our paper and separated these experiments. The self-supervised ImageNet-100 classification result is in **Table 5** and the corresponding object detection result is in **Table 6**.
>
> ---
>
> **Q6** The phrase "defense for detection or detection" is hard to understand.
>
> **A6** This is a typo. It should be "defense for object detection or segmentation" and we have corrected this typo in the revision.
>
> [1] Wang et al., Neural cleanse: Identifying and mitigating backdoor attacks in neural networks. 2019.
>
> [2] Shan et al., Gotta catch'em all: Using honeypots to catch adversarial attacks on neural networks. 2020.
>
> [3] Weng et al., On the trade-off between adversarial and backdoor robustness. 2020.
>
> ---
> Thanks for your constructive comments. Hope our explanations and additional experiments can address your concerns. Please let us know if there is more to clarify. We are happy to take your further questions during the rebuttal stage.

---

> ### Author Response · Authors · 2022-11-18
> **Response to Reviewer eqcZ (1/2)**
>
>
> We thank Reviewer eqcZ for your detailed reviews and appreciation of our work. We fixed our typos and will address the points that you are concerned about below.
>
> ___
>
> **Q1** The connection between adversarial and backdoor examples is not discussed.
>
> **A1:** First, in the feature space of backdoored backbones, we find that the cluster of adversarial samples is really close to the cluster of backdoored samples, which suggests that we are able to use adversarial samples to mimic the behaviors of backdoored samples. Intuitively, it is easy to understand. When crafting adversarial examples, we attempt to maximize the classification with a minor change in the input. Considering the backdoored model (models trained on the poisoned dataset) has an extremely easy "shortcut" in its decision boundary (a small trigger is enough to change outputs totally) adversarial perturbations probably make use of it.
>
> ---
>
> **Q2** Why can we mitigate the backdoor by pulling adversarial images toward its benign parts?
>
> **A2:** In Figure 1(a) and 1(c), for a backdoored backbone, once we add a  trigger pattern into the image, the image will move into a separate cluster (black dots) rather than the original cluster of its ground-truth label, which indicates the backdoor model has a "shortcut" with which a small trigger is enough to change outputs totally. To remove the effect of such a "shortcut", we should pull backdoored samples to their original cluster. Unfortunately, the defender never knows the backdoor trigger and is unable to pull backdoored samples directly. In this paper, we show that the cluster of adversarial data (gray dots) is quite close to the cluster of backdoor data. Thus, we recommend pulling adversarial images, rather than unknown backdoored images, towards their original cluster.
>
> ---
>
> **Q3** More datasets should be used.
>
> **A3** We present more experiments with ImageNet-100 dataset on both supervised and self-supervised backbones using Blend Attack and BadNets. Like the Cifar-10 experiments, we compare our proposed CBD with other supervised defense baselines on supervised backbones and only compare our method with standard contrastive fine-tuning as the label is inaccessible in real-world self-supervised learning. As shown in the following table, it is clear that our method reaches the highest performance among all other baselines. More details can be found in  **Section 4.1**.
>
> > **Supervised backbones**
> >
> > | Attacks | Metrics | No Defense | ANP   | FP    | FT    | NAD   | CBD       |
> > | ------- | ------- | ---------- | ----- | ----- | ----- | ----- | --------- |
> > | BadNets | ACC     | 78.15      | 65.14 | 50.00 | 61.40 | 55.08 | **66.43** |
> > |         | ASR     | 99.90      | 49.45 | 19.88 | 97.35 | 80.08 | **13.39** |
> > |         | PA      | 0.10       | 41.23 | 28.97 | 85.59 | 13.21 | **59.84** |
> > | Blend   | ACC     | 80.06      | 69.42 | 50.08 | 61.48 | 56.62 | **75.39** |
> > |         | ASR     | 99.48      | 18.16 | 18.38 | 40.10 | 26.61 | **0.12**  |
> > |         | PA      | 0.40       | 27.66 | 14.81 | 21.15 | 16.97 | **70.77** |
>
> > **Self-supervised backbones**
> >
> > | Defenses   |       | BadNets |       |       | Blend |       |
> > | ---------- | ----- | ------- | ----- | ----- | ----- | ----- |
> > |            | Acc   | ASR     | PA    | Acc   | ASR   | PA    |
> > | No Defense | 61.52 | 38.56   | 35.45 | 59.86 | 0.02  | 40.06 |
> > | FT         | 38.22 | 5.24    | 32.63 | 34.72 | 0.58  | 21.82 |
> > | Our method | 57.42 | 9.57    | 49.84 | 56.39 | 0.44  | 54.36 |
>
> ---
> (more below)

---

### Official Review · Reviewer_4Caq · 2022-10-25

**Confidence:** 4
**Correctness:** 2
**Technical Novelty And Significance:** 3
**Empirical Novelty And Significance:** 2
**Recommendation:** 3

**Clarity, Quality, Novelty And Reproducibility:**

Clarity: I find myself struggle to follow the paper. The motivation is unclear and the methodology part in Section 3.3 is also very focused. Several terms are not well-defined and the overall setting is vague.

Quality: The paper organization could be improved. The motivation should include why the self-supervised loss would be used and knowledge distillation is needed. Also, the writing could be improved as well.

Novelty: The proposed setting is interesting and could be a useful direction to explored. However, I am not sure what particular model they focus in the paper so it affects the understanding on the paper’s novelty.

Reproducibility: All hyperparameters are listed and detailed. I think the proposed work doesn’t have reproducibility problem.


**Strength And Weaknesses:**

Pros:
1.	The idea of using self-supervised method to defend against backdoor attack is interesting.
2.	The proposed method shows it could both defend against backdoor attack in the self-supervised setting and achieve similar performance in the supervised setting.

Cons:
1.	The paper is not well-organized and hard to follow. The motivation in Figure 1 is not clear and not illustrated. It only mentions the Figure 1a and 1c shows the adversarial example and backdoor examples are close in this projection. However, I am not sure what the difference between different scenarios in a,b,c,d. Also, I find myself couldn’t follow the section 3.3 as well. I don’t see why the contrastive loss would be better than the supervised loss and why the knowledge distillation is necessary.
2.	The overall setting of the proposed method is vague. The paper aims to provide a method to purify the backdoored model without providing the label. However, it is not clear to me model is a self-supervised model or could be a supervised model as well. It then becomes very confused to interpolate the Table 2 as the pseudo-label data. What’s the pseudo-label’s definition? Why do we need to do the pseudo-label experiments?
3.	The performance improvement is also limited. In Table 1 and 2, the performance is similar with the previous baselines. Therefore, is the main contribution of the paper to improve the efficiency or to provide a way to do it in a self-supervised manner?


**Summary Of The Paper:**

The paper proposes a self-supervised method to defend against backdoor attacks. Specifically, motivated by the adversarial example and backdoor example are closed in the feature space, the paper proposes to pull adversarial examples toward its benign parts by constructing a normalized temperature-scales cross entropy loss. Further, it uses the knowledge distillation to boost the clean example performance. The experiments have been conducted on CIFAR10 dataset in both supervised and self-supervised setting. The results show the proposed method could achieve similar performance with other baselines in supervised setting and be able to defend against backdoor attack in the self-supervised setting.

**Summary Of The Review:**

Overall, the proposed method proposes an interesting setting to defend against backdoor attacks. However, due to the paper's written quality and poor-organization, it is hard to understand the main message from the paper.

---

> ### Author Response · Authors · 2022-11-18
> **Response to Reviewer 4Caq (2/2)**
>
> (continued)
>
> **Q4:**  It is not clear to me whether the model is a self-supervised model or could be a supervised model as well.
>
> **A4:**  In this paper, we attempt to design a self-supervised backdoor defense without any labeling requirement. Since our method only focuses on the feature space, it can be applied to not only supervised backdoored backbones, but also self-supervised backdoor backbones. In our paper, we include both backdoored backbones to show that our method is a general defense.
>
> ---
>
> **Q5:** What’s the definition of the pseudo-label? Why do we need to do the pseudo-label experiments?
>
> **A5:** In our paper, the "pseudo-label" of a sample indicates the class which has the highest predicted probability by the supervised backbone on that sample. The previous defense methods always required the labels of clean data, making it impossible to apply to the situation using only unlabeled data (the case studied in our paper). To compare our method with previous studies with the same amount of knowledge (only unlabeled data), we modify these supervised defenses by using the predicted "pseudo-label" as the label for clean data to defend against backdoor attacks. Note that this situation is only possible if the backdoored backbones are trained in a supervised manner. More details can be found in Section 4.1 of the revision.
>
> ---
>
> **Q6** Is the main contribution of the paper to improve efficiency or to provide a way to do it in a self-supervised manner?
>
> **A6** Our main contribution is to define a more practical situation and find a solution to address this more general setup. Although numerous defense methods were proposed to tackle backdoor attacks, most of them mainly focus on the setting where they can use classification loss and costly labeled data, which ignore the limited ability of users that choose to use third-party pre-trained backbones. In this paper, we explore a new setting where the user might only obtain a pre-trained backbone and cheap unlabeled data. Also, without any labeled data and assumption of pre-trained backbones, we can achieve comparable results with the state-of-the-art and even much better results on our ImageNet-100 experiments in updated **Section 4.1**.
>
> [1] Shan et al., Gotta catch'em all: Using honeypots to catch adversarial attacks on neural networks. 2020.
>
> [2] Weng et al., On the trade-off between adversarial and backdoor robustness. 2020.
>
> ---
> Thanks for your constructive comments. Hope our explanations can address your concerns. Please let us know if there is more to clarify. We are happy to take your further questions during the rebuttal stage.

---

> ### Author Response · Authors · 2022-11-18
> **Response to Reviewer 4Caq (1/2)**
>
> We thank Reviewer 4Caq for carefully reading our work and for constructive comments. To alleviate your concerns, we have already updated our new version of the submission to improve the paper writing and organization. Also, we will explain the weakness mentioned in the review.
>
> ___
>
> **Q1:**  The motivation in Figure 1 is not clear and not illustrated.
>
> **A1:**  We added more details in the paper to better understand Figure 1 in **Section 3.2** in the revision. In Figure 1, we study how backdoored data affect backbones by visualizing the locations of different types of data (benign data, backdoored data, and adversarial data) in the feature space on both a supervised backbone in Figure 1(a) and a self-supervised backbone in Figure 1(c). We can easily find: 1) once patched with the trigger pattern, the data will move into a separate cluster (black dots) rather than the original cluster of its ground-truth label, which indicates the backdoor model has a "shortcut" with which a small trigger is enough to change outputs totally; 2) the cluster of adversarial data is quite close to the cluster of backdoor data, which indicates we can use adversarial data to mimic the behavior of backdoored data. Inspired by these findings, we propose to use adversarial data to replace these unknown backdoored data and let these simulated "backdoored data" come back to the original cluster belonging to its ground-truth label. Besides, we also visualize the data locations in the feature space of the purified supervised / self-supervised backbones by our method, to demonstrate the effectiveness of our method.
>
> ---
>
> **Q2** I don’t see why the contrastive loss would be better than the supervised loss.
>
> **A2** We want to clarify some potential misunderstandings of our setting and goal first. Unlike previous post-processing backdoor defense works, which assume the user can obtain both labeled data (costly) and classifier-head (only applicable for supervised backbones), we propose a more practical setting. In particular, we aim to design a backdoor defense that only requires cheap unlabeled data (more friendly for users who need to deploy third parties pre-trained backbones) and can work well on more general backbones (thus applicable for various kinds of backbones, *e.g.*, recent self-supervised backbone). Therefore, without accessing labels, we can only leverage instance-wise contrastive loss instead of supervised loss to do the defense.
>
> ---
>
> **Q3** Why the knowledge distillation is necessary?
>
> **A3** When we attempt to purify the backdoor model, there are actually two goals: 1) keeping the utility (accuracy on benign data) of the original model, and 2) removing the backdoor behaviors. While the Backdoor-to-Standard Pulling Loss aims to remove the backdoor behaviors (Target 2), we still require to apply Embedding Distillation Loss to keep the accuracy on benign data of the original backdoored data (Target 1). Further, we conduct an ablation study to show the necessity of knowledge distillation in Table 8 of the revision. If we remove the Embedding Distillation Loss, the natural accuracy has an obvious drop (e.g., from 89.74% to 80.65% under the BadNets attack).
>
> ---
> (more below)

---

### Author Response · Authors · 2022-11-18
**A Summary of Paper Updates**

We thank all reviewers for their constructive suggestions. Following their suggestions, we made following major changes to the paper.
* **Figure 1** Redraw the figure. Add red rectangle and more captions to better understand our approach.
* **Section 3.1** Add more concise Attack and Defense setup.
* **Section 3.2** Rewrite the motivation and discovery for clarity.
* **Section 3.3** Reorganize the order of losses, add more details on Backdoor-to-Standard pulling and Embedding Distillation.
* **Section 4.1** Add analysis of our prposed CBD defense, **new experiments about Clean Label Attack on Cifar-10**, more details on pseudo-label experiments, and **new experiments on ImageNet-100**.

---

### Decision · Program_Chairs · 2023-01-20

**Decision:**

Reject

**Justification For Why Not Higher Score:**

see above

**Justification For Why Not Lower Score:**

N/A

**Metareview: Summary, Strengths And Weaknesses:**

This work proposed a backdoor mitigation method using unlabeled data, based on contrastive learning.

There are several important concerns from reviewers:
1. Very vague setting. For example, the input is a supervised or self-supervised backbone. According to the rebuttal, it should be supervised. But the authors emphasized pre-trained backbone multiple times, giving an impression of self-supervised backbone.
2. Motivation. The connection between backdoored samples and adversarial examples are not well explained, thus the motivation of the proposed method is not clear.
3. The illustration of the method part is not clear. (I read it, with the same feeling)
4. Unsatisfied results compared to other baselines.

Besides, I  have some additional concerns:
1. It is clearly that this work got the main inspiration from DBD (ICLR 2022), including the clustering and separation of backdoored samples (see fig 1), as well as the defense strategy using contrastive learning. But the authors didn't mention these aspects, instead emphasizing that DBD required labeled data. It is misleading, as DBD is a secure training defense, where the input is a poisoned dataset, rather than a backdoored model. The setting between DBD and this work is different. Besides, even the different setting, DBD could be still compared, if the input is a poisoned dataset, a backdoor model could be firstly trained, then utilizing the proposed CBD method.
2. Since given a supervised trained model, there are backbone and classifier. The CBD only fine-tune the backbone, how about the classifier, unchanged? I doubt the performance if the combined model with a fine-tuned backbone and a fixed classifier.
3. The authors emphasized the unlabeled data is the main difference with existing works. However, what is the relationship between this unlabeled data and the training/testing data? Besides, one ECCV 2022 work "Data-free Backdoor Removal based on Channel Lipschitzness" doesn't require any data.